# Capsid Integrity qPCR—An Azo-Dye Based and Culture-Independent Approach to Estimate Adenovirus Infectivity after Disinfection and in the Aquatic Environment

**Mats Leifels [1,2,\*]**, **David Shoults [2]**, **Alyssa Wiedemeyer [2]**, **Nicholas J. Ashbolt [2]**,
**Emanuele Sozzi [3]**, **Angela Hagemeier [4] and Lars Jurzik [5]**

1    Singapore Centre for Life Science Engineering, Nanyang Technological University,
     Singapore 637551, Singapore
2    School of Public Health, University of Alberta, Edmonton, AB T6G 2G7, Canada; dshoults@ualberta.ca (D.S.);
     wiedemey@ualberta.ca (A.W.); ashbolt@ualberta.ca (N.J.A.)
3    Department of Environmental Science and Engineering, Gilling's School of Global Public Health,
     University of North Carolina, Chapel Hill, NC 27516, USA; esozzi@email.unc.edu
4    Department of Hygiene, Social and Environmental Medicine, Ruhr-University Bochum,
     44801 Bochum, Germany; Angela.Hagemeier@mpi-dortmund.mpg.de
5    Federal State Agency for Consumer & Health Protection Rhineland-Palatinate, 56068 Koblenz, Germany;
     lars.jurzik@lua.rlp.de
\*    Correspondence: mats.leifels@rub.de

**Abstract:** Recreational, reclaimed and drinking source waters worldwide are under increasing anthropogenic pressure, and often contain waterborne enteric bacterial, protozoan, and viral pathogens originating from non-point source fecal contamination. Recently, the capsid integrity (ci)-qPCR, utilizing the azo-dyes propidium monoazide (PMA) or ethidium monoazide (EMA), has been shown to reduce false-positive signals under laboratory conditions as well as in food safety applications, thus improving the qPCR estimation of virions of public health significance. The compatibility of two widely used human adenovirus (HAdV) qPCR protocols was evaluated with the addition of a PMA/EMA pretreatment using a range of spiked and environmental samples. Stock suspensions of HAdV were inactivated using heat, UV, and chlorine before being quantified by cell culture, qPCR, and ci-qPCR. Apparent inactivation of virions was detected for heat and chlorine treated HAdV while there was no significant difference between ci-qPCR and qPCR protocols after disinfection by UV. In a follow-up comparative analysis under more complex matrix conditions, 51 surface and 24 wastewater samples pre/post UV treatment were assessed for enteric waterborne HAdV to evaluate the ability of ci-qPCR to reduce the number of false-positive results when compared to conventional qPCR and cell culture. Azo-dye pretreatment of non-UV inactivated samples was shown to improve the ability of molecular HAdV quantification by reducing signals from virions with an accessible genome, thereby increasing the relevance of qPCR results for public health purposes, particularly suited to resource-limited low and middle-income settings.

**Keywords:** capsid integrity qPCR; human adenovirus; surface water; viral infectivity; water quality indicator

## 1. Introduction

Wastewater treatment technology has markedly improved in efficacy over the past decades [1]. Waterborne diseases caused by human or animal fecal contamination, however, remain a threat

to public health worldwide [2–4]. Traditionally, the quality of surface water used for recreational activities and drinking water production relies on fecal bacterial indicators such as *Escherichia coli* and enterococci [5,6]. Utilizing bacteriophages or enteric viruses of fecal origin instead has been proposed for some time [7–9], and a recent study could demonstrate that bacteria show substantially lower resistance towards environmental stressors and disinfectants such as UV-light and ozone when compared to enteric viruses [10]. Furthermore, human enteric viruses tend to drive infection risk for exposures to waterborne pathogens [11–15]. Hence, there is an increasing focus from government regulators and scientists to identify useful viral indicators [16–18]. While nearly 150 waterborne enteric viruses have been associated with human illnesses [19], human adenoviruses (HAdV) are considered to be one of the most appropriate indicators for human fecal viral presence in waters [7]. The suitability of HAdV derives from their environmental resistance and relative abundance year-round (unlike Rotavirus or Enterovirus), their well-established cell-culture infectivity assays to determine infectivity, their morphological similarity to more critical human pathogenic viruses such as Rotavirus and Norovirus GII [7,20–23] and the lower risk of infection for the handling researcher. Unlike bacteriophages, which have been shown to at least be theoretically able to replicate in the presence of their bacterial hosts in environmental biofilms and come from non-human sources [8,24,25], HAdV provides specificity to the human host. Preferred viral sample processing has been reviewed by [7,26], and in brief consists of primary virus concentration by filtration or non-filtration-based techniques, and then estimation of adenoviruses by quantitative polymerase chain reaction (qPCR) or cell-culture methods. Molecular qPCR is cost-efficient, robust, and specific. However, it is not able to differentiate between infectious and non-infectious viral particles, and so includes what may be considered false positive signals from a public health viewpoint [27,28]. Therefore, methods including cell culture continue as the gold standard despite the high cost and skill level required. Additionally, elaborate and mandatory safety facilities are necessary, which pose a barrier to its implementation for environmental monitoring in low (as well as in high) income countries [29]. These described limitations make the development of alternative and preferably culture-independent methods to estimate virus infectivity a prime target of interest.

However, the issue of overestimating virus concentration by widely used quantitative PCR (qPCR) has been the topic of several publications over the past decade [30–32]. According to results from both laboratory (low complexity water matrix) and real-life samples (higher complexity water matrix), the implementation of azo-dye pretreatment using both ethidium monoazide (EMA) and propidium monoazide (PMA) may improve the capabilities of already established and routinely used qPCRs—especially for HAdV—to remove "false-positive" signals from samples in complex environmental settings such as surface water and wastewater [33,34].

In the past decade, capsid integrity (ci)-qPCR, a technique derived from the so-called viability qPCR first used by [35] to estimate bacterial viability, has been considered as a potential modification to routinely used qPCR assays to enhance the value from qPCR assays [36]. In the context of environmental virology and the monitoring of food items for enteric viruses, this technique could help improving widely used detection methods while not requiring extra equipment [33], especially for Norovirus GI and GII for which culture-based detection methods are not available [37–39]. Prior to molecular quantification, concentrated water samples are treated with the virus capsid-impermeable and genome-intercalating azo-dyes ethidium monoazide (EMA) or propidium monoazide (PMA). The pretreatment has been performed for disinfection studies and in environmental virology, with general success in the removal of "false positive" qPCR results [40–45]. The mode of amplification inhibition by EMA and PMA dyes has been described in detail by [35]. For enteric viruses, however, both viral concentration and co-concentrated environmental inhibitory substances may limit the applicability of ci-qPCR [42,46].

The efficiency of the azo-dye pretreatment in bacteria is influenced by certain characteristics of the qPCR target region and the formed amplicon due to differences in length, secondary structure, and ratio of nucleotides [15,47,48]. Therefore, the current work was undertaken to assess two of the most

commonly used HAdV qPCR protocols [49,50] for their compatibility with azo-dye pretreatment to detect infectious HAdV. This was performed before and after inactivation by heat, chlorine, or UV light in a laboratory scale. Further objectives were to describe any differences in EMA or PMA to remove non-infectious HAdV virions in complex water matrices (51 environmental and 24 wastewater samples before and after treatment in an UV pilot facility) using ci-qPCR and to compare the results against cell-culture assays.

## 2. Material and Methods

### 2.1. Propagation and Enumeration of Virus Stocks

HAdV type 5 (HAdV5, provided by the Department of Molecular Virology at Ruhr-University Bochum, Bochum, Germany) was propagated on A549, a human lung adenocarcinoma cell line, in Dulbecco modified eagle medium (DMEM) (Sigma-Aldrich, St. Louis, MO, USA) supplemented with 10% fetal bovine serum (FBS) and 1% penicillin-streptomycin incubated at 37 °C with 5% $CO_2$. HAdV stock was suspended in DMEM added with 20% glycerol to minimize freezing damage before being aliquoted and stored at −80 °C until used for the study. A549 cells were pipetted into 48-well cell culture plates to approximately $5 \times 10^4$ cells per well, centrifuged (6000 g for 5 min at room temperature) and washed 3 times with phosphate-buffered saline (PBS). Cells were infected with 50 µL of virus stock solution, serially diluted in sterile filtrated PBS, and incubated for 90 min at 37 °C and 5% $CO_2$, then followed by the addition of 200 µL of maintenance medium with 2.5% FBS. Plates were incubated for 4–6 days and monitored for cell de-attachment and cytopathic effects. Following incubation, cells were fixed with 10% formaldehyde and stained with crystal violet solution (Merck, Darmstadt, Germany) to estimate the tissue culture infectivity dose (TCID$_{50}$) of HAdV5. Approximately $5.0 \times 10^6$ genomic copies of HAdV5 have been added to 1 mL of PBS for each inactivation method (eight replicates per disinfection). Murine norovirus (MNV) provided by the Friedrich Loeffler Institute, Greifswald-Insel Riems in Germany, was propagated in RAW264.7 cells (ECACC). RAW264.7 cells were cultured in $1 \times$ RPMI1640 medium supplemented with 10% FBS. Virus titers have been calculated following the protocol for HAdV.

### 2.2. Thermal, UV, and Hypochlorite Inactivation

Virus inactivation was conducted in accordance to [51] using UV dose fluences of $150 \pm 17.5$ mJ·cm$^{-2}$ UV-C (low-pressure 25 W germicidal UV lamp emitting light at a wavelength of 253.7 nm; Phillips, Hamburg, Germany) for monochromatic UV inactivation. Heat inactivation of HAdV stock was performed at 95 °C for 10 min in reaction tubes using a heat block. For chemical disinfection, HAdV was diluted in chlorine-demand free water with a consistent concentration of 2 mg·L$^{-1}$ of available free chlorine (generated from hypochlorite, Merck, Germany) incubated at room temperature for 2 min, then adding 50 µL of 10% thiosulfate to quench remaining chlorine. The absence of free and reactive chlorine was verified using free and total chlorine strips (0–10 mg Cl$_2$ Free and Total Chlorine Strips, Hach Lange, Germany).

### 2.3. Collection and Concentration of Water Samples for Virus Analysis

The environmental and wastewater samples analyzed in the second part of this study were collected between 20 May and 10 September 2015 (bathing season as proposed by the Safe Ruhr Project, conducted by the German Federal Ministry for Research and Education) as described in [33,52]. River and UV treatment influx/efflux samples of 10 L volume were taken bi-weekly from the urban Ruhr River (51 samples) as well as the late stages of a sewage water treatment facility (a total of 24 samples, 12 before and 12 after UV treatment) in Essen, Germany. Physio-chemical, microbiological characteristics, and sampling coordinates were previously described by [52]. After transportation to the laboratory on ice and within a 3 h limit, samples were concentrated using the optimized virus absorption and elution protocol first described by [53] and adapted for the river under investigation

by [54]. In brief, MgCl$_2$ was added to a final concentration of 0.05 M to stabilize the viral capsids. Subsequently, pH was adjusted to 3.5 and a vacuum pump used to pass the sample through a negatively charged nitrocellulose membrane (Merck, Darmstadt, Germany) with 0.45 μm pore size resulting in viral adsorption. Elution and recovery of viruses were conducted by incubating and eluting the filter with a non-organic elution buffer (0.005 M KH$_2$PO$_4$, 0.5 M NaCl, 0.1% (v/v) Triton x-100, and pH 9.2). Secondary concentration was performed with 12.5% polyethylene glycol 6000 (PEG) and 2.5% NaCl for 3 h at 4 °C before centrifugation of the sample at approximately 15,000 g, discarding of the supernatant and resuspension of the pellet in PBS. Concentrated samples were stored at −80 °C until analysis by molecular or culture-based methods. Each sample was spiked with non-endemic murine norovirus (MNV) right after arrival in the laboratory as a non-competitive internal control [55]. Quantification of this virus showed that total virus recovery rates between 50% and 75% (data not shown) could be achieved for the filtration process and genome extraction.

### 2.4. Dye Pretreatment

Sample aliquots (200 μL) of infectious and inactivated HAdV were treated with EMA or PMA (Biotium Inc., Hayward, CA, USA) according to [46,51] using a final concentration of 0.04 mM. Tubes were then mixed gently by inverting several times then incubated on ice for 30 min before being transferred to the LED-based PhaST Blue System (IUL, Barcelona, Spain) and photo-activated at 100% light intensity for 15 min. To determine the interference of both dyes with the detection ability of qPCR, negative controls were included with each set of reactions by treating viruses with either EMA or PMA without photo-activation. Furthermore, extracted HAdV5 naked nucleic acids were exposed to the same concentrations of azo-dyes in serial dilutions to ensure the binding capacity of both EMA and PMA was adequate (mean removal of >10$^6$ genomic copies per 20 μL; data not shown). The HAdV protocol combined with the EMA/PMA pretreatment was compared to standalone qPCR and cell culture to assess their ability to eliminate the detection of HAdV with an accessible genome (presumed non-infectious) from total virion counts.

### 2.5. Extraction of Viral DNA and Quantification of Adenovirus Genomes

Whole HAdV genomes were extracted using the QIAmp DNA Blood Mini Kit (Qiagen, Hilden, Germany) following the manufacturer's protocol, eluted in 100 μL of the provided inorganic elution buffer, then stored at −20 °C until analyzed. All primers and probes used, and amplicon lengths are listed in Table 1. Genome standards were produced according to the cloning protocol described in detail by [54] and cDNA synthesis for MNV RNA was performed using the High Capacity cDNA Reverse Transcription Kit (Thermo Fisher, Waltham, MA, USA) following the manufacturer's protocol. In brief, PCR amplicons for HAdV and MNV were generated with primers and probes listed above. Amplicons of the target region under investigation were then cloned into the PCR 2.1 vector (Thermo Fisher, Waltham, MA, USA), following the manufacturers protocol before being purified with the QIAprep Spin Miniprep Kit (Qiagen, Hilden, Germany). Concentrations of the purified plasmid were determined by using the Qubit 2.0 Fluorometer (Thermo Fisher, Waltham, MA, USA). The number of molecules per microliter of standard RNA/DNA was then calculated from its molecular weight and the length of the sequence. For the qPCR assay, the Takyon no ROX qPCR Mastermix (Eurogentec, Liège, Belgium) was used in a reaction volume of 20 μL containing 5 μL (2 μL of cDNA) of nucleic acid template, 0.25 μM of both forward and reverse primers, and 1 μM of specific probe. Cycling conditions for HAdV-Heim were as follows: Uracil-N-glycosidase (UNG) activation at 50 °C for 2 min (to remove carry-over contamination), 95 °C for 3 min, then 45 cycles of 95°C for 15 s, 60 °C for 1 min, and 78 °C for 5 s to acquire the fluorescence. Conditions for HAdV-Hernroth were as follows: UNG activation at 50 °C for 2 min, 94 °C for 3 min, then 40 cycles of 95 °C for 15 s, and 60 °C for 60 s to acquire the fluorescence. Conditions for MNV are as follows: cDNA synthesis for 120 min at 37 °C followed by UNG activation at 50 °C for 2 min, 94 °C for 3 min and 45 cycles of 95 °C for 15 s and 60 °C for 60 s. Acquisition of fluorescence was performed at 80 °C for 5 s. For all viruses, the Rotorgene 6000 cycler

system (Qiagen, Hilden, Germany) was used for PCR amplification. The limit of detection (LD) in copies per qPCR reaction was determined via serial dilution of the known standard and calculated to be 25 genomic copies. Calculated by the dilution factors of the filtration and concentration process, the lowest number of genome copies has been determined to be 100 copies per liter for HAdV and MNV. The number of replicates for each inactivation condition and protocol was eight (treatment conducted four times in duplicates).

**Table 1.** Primer and probe sets used for qPCR detection of human adenoviruses (HAdV) and murine norovirus (MNV). Wobble bases: W = A/T; R = A/G; Y = C/T; K = G/T; and S = C/G.

| Protocol | Primer | Primers & Probe Sequences 5′–3′ | Amplicon (bp) | Gene Target | Ref. |
|---|---|---|---|---|---|
| HAdV-Heim | AQ1 | GCC ACG GTG GGG TTT CTA AAC TT | 139 | | [50] |
| | AQ2 | GCC CCA GTG GTC TTA CAT GCA CAT C | | | |
| | AdV-P | [Hex]-TGC ACC AGA CCC GGG CTC AGG TAC TCC GA-[BHQ1] | | Hexon | |
| HAdV-Hernroth | Ad.hex.up | CWT ACA TGC ACA TCK CSG G | 69 | | [49] |
| | Ad.hex.do | CRC GGG CRA AYT GCA CCA G | | | |
| | AdV-ACDEF | [6FAM]-CCG GGC TCA GGT ACT CCG AGG CGT CCT-[TAMRA] | | | |
| | AdV-B | [6FAM]-CCG GAC TCA GGT ACT CCG AAG CAT CCT-[TAMRA] | | | |
| MNV | TMP 1 | AGA GGA ATC TAT GCG CCT GG | 92 | ORF2 | [56] |
| | TMP 2 | GAA GGC GGC CAG AGA CCA C | | | |
| | TMP | [6FAM]-GCC ACT CCG CAC AAA CAG CCC-[BHQ1] | | | |

## 3. Statistical Analysis

Data analysis of both laboratory scale and environmental samples was conducted using Microsoft Excel 2016. Pearson product-moment correlation coefficients were calculated for virus quantity data obtained by qPCR, PMA-ci-qPCR, EMA-ci-qPCR, and cell culture. All *p*-values below 0.05 were determined to be statistically significant.

## 4. Results and Discussion

Although HAdV-Hernroth amplicons are only half the length of HAdV-Heim (which should statistically influence the number of azo-dye molecules bound to free genomic material), the ability of PMA and EMA pretreatment to remove non-infectious virions ("false-positive" results) was observed to be similar. The ci-qPCR estimates using either dye resulted in a reproducible reduction in targeted gene presence when HAdV was inactivated by chlorine or heat treatment, compared to total virions by qPCR. However, no notable signal reduction was observed after UV exposure, which has been described before [43,51]. Therefore, the utilization of either dye—with slightly better rates for PMA—presumably removed "false-positive" signals from non-infectious virions when loss of infectivity is correlated with the integrity of the virus capsid. The failure of both dyes to estimate infectious HAdV after UV disinfection is consistent with studies with PMA/EMA-qPCR for UV-treated bacterial assays [35,57,58] and most likely due to the different mode of virus inactivation (UV breaking down di-hydrogen bonds between nucleotides rather than affecting the virus capsid as described by Beck et al. [10]). As for bacteria, similar observations have been described for a variety of enteric viruses (such as MNV, Poliovirus, Norwalk virus, Rotavirus, HAdV, and Coxsackievirus B1) and bacteriophages (like MS2 and PhiX174) after exposure to UV light by [38,59–61]. While culture methods indicated a loss of virus and phage infectivity, the addition of PMA and/or EMA failed to represent this. Importantly for the presented work, though, no loss of signal due to EMA/PMA addition to the sample under laboratory inactivation conditions was detected using qPCR and it can be assumed that adding the pretreatment step does not have a negative effect on qPCR sensitivity. Cell culture assays indicated

that all inactivation methods used (UV, heat, and chlorination) resulted in the complete loss of HAdV infectivity (see Figure 1A–C), although each method relies on different mechanisms of virus inactivation.

As a highly reactive substance, free chlorine affects both the adenoviral genome and capsid [62] and, as expected, its usage allowed the dyes to enter the virus and interact with the virus genome. This is in accordance with other studies, which investigated this disinfection method before utilization of ci-qPCR for different enteric viruses. While qPCR showed little to no signal reduction after 2 min of 2 mg/mL of free chlorine for HAdV, Enterovirus, Rotavirus and the phages MS2 and PhiX174, pretreatment with PMA and EMA indicated a loss comparable to cell culture in our previous work [51]. Similar observations have been published by [63–66] for several members of the Enterovirus group, MS2, and Aichivirus 1, as well as for Noroviruses GI/GII and an Astrovirus after their exposure to different concentrations of hypochlorite and free chlorine. Higher temperatures, which are known to denaturate the virus capsid, showed comparable results when analyzed with azo-dyes. Articles published by [34,67–69] indicate that inactivation of Noroviruses GI/GII and the Hepatitis A virus, as well as Rotavirus and Aichivirus 1 after inactivation at temperatures between 50 °C and 95 °C was observable when PMA derived PMAxx, PMA, and EMA pretreatment was applied.

As shown in Figure 1A,B, qPCR using both HAdV protocols resulted in no significant $\log_{10}$ reduction following inactivation of all adenoviral suspensions. The cell culture assays strongly indicate that chlorine and temperature treatment led to a complete loss of adenovirus infectivity ($\geq 4.34 \log_{10}$; see Figure 1A,C) and UV to a large loss ($3.46 \log_{10}$; Figure 1B). These results demonstrate that relying on qPCR alone would overestimate the amount of infectious HAdV, with reduction rates of merely $0.07 \log_{10}$ after inactivation by UV (HAdV-Hernroth) and $1.07 \log_{10}$ by Chlorine (HAdV-Heim), thus resulting in misleading quantitative microbial risk assessment (QMRA) [26,70]. Utilization of PMA ci-qPCR led to a signal reduction that corresponded in part with cell culture (temperature, HAdV-Heim). However, as discussed above it failed to indicate inactivation by UV treatment. Optimization experiments for *Legionella pneumophila* conducted by [71] showed similar trends and indicated that azo-dye incubation conditions such as temperature and concentration might have a bigger influence on the ability of PMA and EMA to remove genome accessible pathogens from qPCR reactions and that extending the amplicon region can result in undesired increase of the limit of detection.

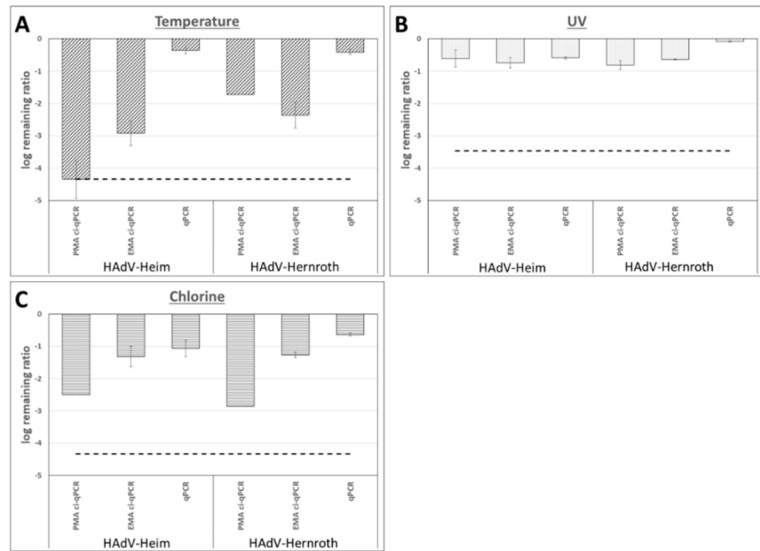

**Figure 1.** $\log_{10}$ remaining ratio of HAdV-Heim and HAdV-Hernroth mean values with and without ethidium monoazide (EMA) or propidium monoazide (PMA) dye pretreatment after exposure of the samples to 95 °C for 10 min (**A**), 150 mJ·cm$^{-2}$ UV-C (**B**) and 4 mg min·L$^{-1}$ free chlorine (**C**). The dotted line represents the results from cell culture and the error bars show the standard deviation. n = 8 for each inactivation condition and the control.

*Correlation between qPCR, ci-qPCR and Cell Culture in Environmental and Spike Samples*

When comparing the results after all EMA/PMA ci-qPCR runs, the estimated concentrations of positive viruses after PMA pretreatment and cell culture was consistently—but not statistically—lower than those after EMA pretreatment (P = 0.26 for PMA and P = 0.24 for EMA). It has to be mentioned, though, that comparison between genome copies per liter and TCID$_{50}$ per liter is inherently problematic but are a reality in environmental virology and certain correlations have been shown by Kuchipudi et al. [72] for influenza viruses and Ryu et al. [73] for HAdV. Consequently, and resulting from the presented work, our preference is the utilization of PMA pretreatment over EMA based on environmental and virus stocks inactivated under laboratory conditions as implied from Figure 1. Possibly higher efficacy of PMA over EMA has been discussed in detail by [74] for *Legionella* spp. in hospital water samples and by Kim et al. [75] for viruses.

Assuming the concentration obtained by cell-culture assays realistically estimated infectious virions [76], PMA/EMA pretreatment of both surface and sewage water DNA samples lead to a reduction in what we refer to as "false-positive" samples when compared to qPCR alone (see Figure 2). While quantification using EMA/PMA ci-qPCR still resulted in the detection of higher virus concentrations than the cell-culture based assay, utilization of the azo-dyes helped narrow the gap between the molecular to culture-based quantification results, something which could not be seen for conventional qPCR and could be beneficial for a more realistic QMRA (see Figure 3). One explanation for the remaining "false-positives" could be through sunlight exposure of environmental virions [77] and routine wastewater disinfection (UV-C 254 nm) from the wastewater treatment facility under investigation [78], both of which are known to leave the virus capsid intact [10], as also discussed above for the current study.

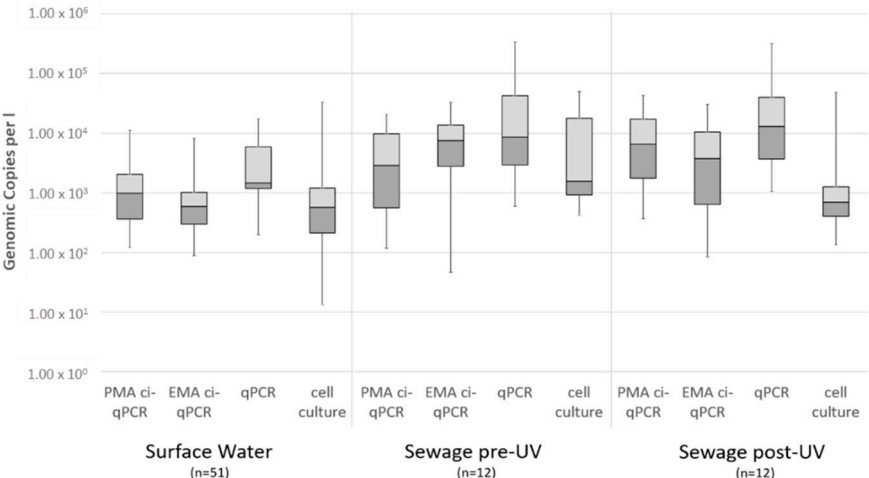

**Figure 2.** Concentration of HAdV in surface water as well as pre- and post-UV treatment, obtained by ci-qPCR using either EMA or PMA followed by qPCR (using the Heim et al. [50] protocol, stated as gene copies per liter of sample) and cell culture (as TCID$_{50}$ per liter). 25- and 75-percentile, median value as well as minimum and maximum; error bars represent the standard deviation. Limit of detection has been calculated as 25 genomic copies per L of sample for molecular methods and 2 infectious virions per mL for the cell.

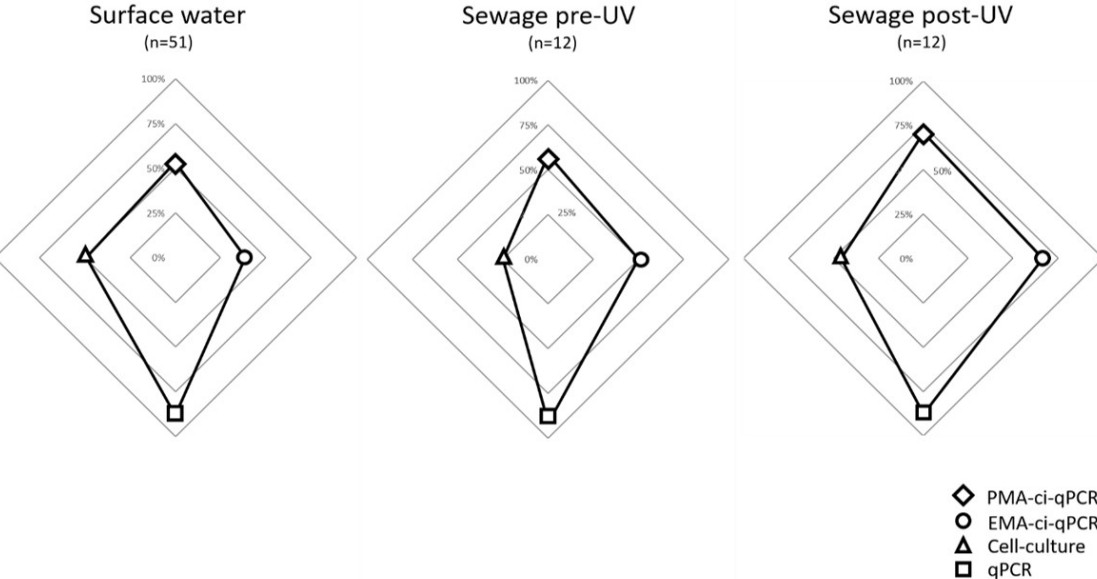

**Figure 3.** Comparison of the percentages of samples positive for HAdV obtained by azo-dye pretreated ci-qPCR, qPCR, and cell culture for the surface and municipally treated wastewater before and after UV-treatment in the pilot plant. Limit of detection has been calculated as 25 genomic copies per L of sample for molecular methods and 2 infectious virions per mL for the cell culture.

Aside from removing a fraction of presumably non-infectious ("false-positive") quantified virions, the application of pretreatment using either azo-dye was successful for all sample types under investigation. Of the 51 surface water and 24 wastewater samples (12 before and 12 after UV treatment), over 90% were measured positive by qPCR alone and the estimated HAdV concentration was higher than the limit of detection (Figure 2). After pretreatment with EMA or PMA, the ci-qPCR exhibited quantification of HAdV in approximately 50% of the samples (53% for PMA in surface water and 42% before UV treatment) while the cell-culture based assay exhibited quantification in a similar number of samples (see Figure 4; 49% in surface and 51% after UV treatment). Paired comparison for the different modes of virus quantification showed weak significance for both dyes compared to qPCR alone ($p = 0.26$ for PMA and $p = 0.24$ for EMA). It should be noted that those molecular detection methods using PMA/EMA showed higher percentages of positive reads after UV treatment while qPCR alone failed to show this trend. This trend might eventually be explained with a slight reduction of inhibitory substances during the UV exposure and the inherently higher resistance of HAdV towards UV light. Concentration and occurrence of both Enterovirus and Rotavirus measured using ci-qPCR indicated a loss of capsid integrity after UV in previous studies [51].

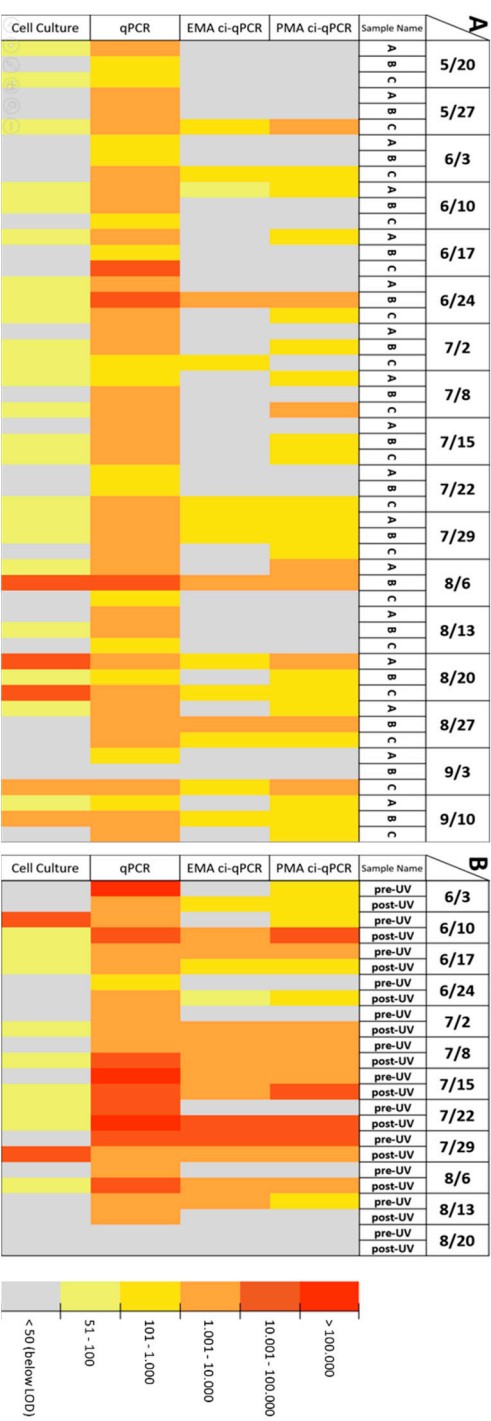

**Figure 4.** (**A**) Concentrations of HAdV obtained by qPCR, cell culture, EMA ci-qPCR, and PMA ci-qPCR between May and September 2015 from the sampled bathing site on the River Ruhr, Essen, Germany. Molecular detection with no signal is indicated as half the limit of detection. (**B**) Concentrations of HAdV obtained by qPCR, cell culture, EMA-ci-qPCR and PMA-ci-qPCR between June and August 2015 before (pre-) and after (post-) UV treatment at the wastewater treatment facility Essen-Süd in Essen, Germany. Molecular detection with no signal is indicated with results lower than half the detection limit (LOD) of 25 copies per liter for molecular detection and 2 infectious virions per liter for cell culture. Values as gene copies per liter per sample.

The spatial–temporal analysis of water samples taken between May and September 2015 (for surface water; May and August in the case of sewage) shows similar results. Hence, the application of ci-qPCR

appears to provide a more realistic representation of adenoviral-positive samples for management and follow-up QMRA applications to assess impacts at various exposure points. As shown in Table 2, the rate of false-positive signals according to the gold standard of cell culture could be reduced massively from 76.47% down to 3.89% for PMA and 7.84% for EMA for surface water. The tendency of EMA to enter viable bacteria and virus cells [79] and thus lead to a suppression of qPCR signals from actually capsid-intact HAdV could explain the difference in false-negative signals between PMA and EMA: While the propidium-derived dye resulted in 9.80% of false-negatives, a third of all reads must be assumed false-negative after application of ethidium monoazide.

**Table 2.** Reduction of presumably false-positive signals from conventional qPCR and EMA/PMA ci-qPCR compared to cell-culture based (representing 100%) assays for surface (n = 51) and treated wastewater (combined pre- and post-UV; n = 24) samples.

| False +/− for Detection of Infectious HAdV | | | |
|---|---|---|---|
| **Source** | **Assay** | **False +** | **False −** |
| Surface (n = 51) | PMA ci-qPCR | 3 (5.89) | 5 (9.80) |
|  | EMA ci-qPCR | 4 (7.84) | 17 (33.33) |
|  | qPCR | 39 (76.47) | 0 (0.00) |
| Before UV (n = 12) | PMA ci-qPCR | 5 (41.67) | 1 (16.67) |
|  | EMA ci-qPCR | 4 (33.33) | 1 (16.67) |
|  | qPCR | 8 (66.67) | 0 (0.00) |
| After UV (n = 12) | PMA ci-qPCR | 2 (16.67) | 1 (8.33) |
|  | EMA ci-qPCR | 2 (16.67) | 2 (16.67) |
|  | qPCR | 8 (66.67) | 3 (20.00) |

## 5. Conclusions

For researchers in lower resource settings without access to enteric virus cell-culture assays, there seems to be considerable value in utilizing ci-qPCR to estimate the concentration of potentially infectious enteric viruses but noting the inability to assay UV-treated samples. Given the generally high persistence of viral pathogens both in the aquatic [46,54] and wastewater environments [80–83], and that enteric virus infection risks to dominate in wastewater reuse applications [84], e.g., for irrigation purposes, the removal of "false-positive" infectious virus results has the potential to increase the significance of qPCR results for public health, economic, and QMRA purposes.

While we demonstrated the utility of adapting two commonly used HAdV qPCR assays, the application of azo-dye pretreatment to established molecular detection techniques for other targets, such as human norovirus (for which cell culture is not routinely available as described by [85]), has considerable potential, as recently described for environmental and food safety applications [34,86,87].

**Author Contributions:** M.L. and J.L. conceived and planned the experiments; M.L. and A.H. carried out the experiments; M.L., E.S. and D.S. contributed to the interpretation of the results as well as conceptualizing and generating the figures; M.L. took the lead in writing the manuscript; D.S., E.S, A.W. and N.J.A. provided much needed critical feedback, helped shape the discussion, analysis and manuscript as a whole; and all authors provided assistance in editing and proofreading the manuscript.

**Funding:** The study was performed within the Fortschrittskolleg FUTURE WATER and funded by the Ministry of Innovation, Science and Research North Rhine-Westphalia, Germany. Parts of the study were conducted within the research project Preventive risk management in drinking water supply (PRiMaT). Financial support by the German Federal Ministry of Education and Research (BMBF) within the framework of its funding program Sustainable Water Management (NaWaM) is gratefully acknowledged (Funding number: 02WRS1279K). The project is part of the funding measure–Risk Management of Emerging Compounds and Pathogens in the Water Cycle (RiSKWa).

**Acknowledgments:** The authors thank Martin Mackowiak for his advice and help.

**Conflicts of Interest:** The authors declare no conflict of interest.

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
