# Peer review of "Capsid Integrity qPCR—An Azo-Dye Based and Culture-Independent Approach to Estimate Adenovirus Infectivity after Disinfection and in the Aquatic Environment"

_water, doi:10.3390/w11061196_

Round 1
Reviewer 1 Report
The manuscript describes the use of commonly known viability dies PMA and EMA in complement to qPCR to try and decrease the number of false-positive samples obtained by qPCR alone with comparison also to the 'gold standard' cell culture. This is a very interesting topic, although several publications have already been performed in it. The main advantage of this manuscript is the usage of several environmental waters (including UV-treated wastewater samples) to apply the different methods. Already there are some concerns regarding the manuscript, mainly when discussing the results. The discussion would largely benefit if the authors were to compare with previous papers performed for viruses instead of comparing mainly with bacterial results. There are several other publications in viruses such as KArim et al. 2015 "Propidium monoazide reverse transcriptase PCR and RT-qPCR for detecting infectious enterovirus and norovirus" (Journal of Virological Methods
Volume 219, July 2015, Pages 51-61), Leifels et al 2016 "Use
of ethidium monoazide and propidium monoazide to determine viral
infectivity upon inactivation by heat, UV- exposure and chlorine" (International Journal of Hygiene and Environmental Health Volume 218, Issue 8, November 2015, Pages 686-693), Randazzo et al. 2016 "Evaluation of viability PCR performance for assessing norovirus infectivity in fresh-cut vegetables and irrigation water" (Int J Food Microbiol. 2016 Jul 16;229:1-6) and Monteiro and Santos 2018 "Enzymatic and viability RT‐qPCR assays for
evaluation of enterovirus, hepatitis A virus and norovirus inactivation:
Implications for public health risk assessment" (J Appl Microbiol. 2018 Apr;124(4):965-976), which the authors can use to benefit the discussion. there are some other issues with the paper. It is not clear how many samples were tested for each wastewater type, were they 24 samples in total for the wastewater samples regardless of the type or were they 24 for each type? Since the authors explain briefly the concentration of the environmental samples, it is not clear how the secondary concentration with PEG was performed and it is not clear the resuspension buffer, if any was used. The authors should explain, if briefly, the cloning protocol used since they refer to another manuscript. The authors do not describe if they use a one-step or a two-step qPCR for the detection of MNV and if it was a one-step qPCR which kit was applied. On page 7, line 193, the authors state that "Although HAdV-Heim amplicons are only half the length of HAdV-Hernroth" which is disagreement with the numbers presented on table 1 (page 6). the authors should re-write the sentence to be in agreement with Table 1. Is the legend from the Y axis on figure 1 correct? Shouldn't it be Log10 reduction as stated in the figure's legend? Additionally, Figure 3 and 4 appear in the text before figure 2. Additionally, Figure 4 is upside down. It would be relevant to understand why heat and chlorine treatment were able to complete inactivate Adenovirus as measured by cell culture but the same did not happen with UV treatment. Other interesting result that the authors fail to address is the fact that a lower percentage of positive results were obtained when analyzing sewage pre-UV when compared to sewage post-UV by cell culture, PMA and EMA-ci-qPCR although the qPCR results are similar for both types of sewage. The authors should also check if the % of positive samples for PMA on lines 278/279 before UV treatment is correct (45%). On page 7 the authors have two statements lines 193-195 "Although HAdV-Heim amplicons are only half the length of HAdV-Hernroth (which should
statistically influence the number of azo-dye molecules bound to free genomic material), similar differences in the ability to remove non-infectious virions (‘false-positive’ results) were observed." and lines 213 and 215 "Still, they differ in the length and sequence of the targeted area, which most likely influences the likelihood of a successful binding of EMA/PMA and the subsequent formation of the genome dye complex" that seems to contradict each other and the finding the authors have obtained. The authors should rearrange the text. The authors state that the higher number of negative environmental samples obtained with EMA-PCR compared to cell culture and PMA-PCR was possibly due to the entrance of EMA into competent cells. Besides the controls added in the dye pretreatment section, didn't the authors also tested an additional control where they treated infectious viruses with EMA and PMA? the manuscript would highly benefit from this.
Minor comments are:
- infectivity is misspelled on page 2 line 50;
- The authors reuse quantitative PCR and abbreviation on page 2 line 65 when they have used it previously in the text (page 2, lines56-57) and the same occurs on page 4 for MNV that is written in full with abbreviation on line 105 nad again in full with abbreviation on line 138. Remove from the second reference;
- on page 3 line 84, the authors should change assessed to assess;
- on page 4, line 127, physico-chemical is misspelled;
- the sentence on page 7 lines 201 to 204 seems to me missing a link between the first and the second part;
- page 9, line 242 it seems to have an extra and after reference [61];
- page 10, line 273 there seems also to be an extra the or a at the beginning of the sentence;
- page 14, line 317 word detects seems incorrect with the remaining of the sentence.
Author Response
Thank you for your very valuable and informative comments, please find the response to each of the points raised in the following table as well as the revised version of the manuscritp.
The manuscript describes the use of commonly known viability dies PMA and EMA in complement to qPCR to try and decrease the number of false-positive samples obtained by qPCR alone with comparison also to the 'gold standard' cell culture. This is a very interesting topic, although several publications have already been performed in it. The main advantage of this manuscript is the usage of several environmental waters (including UV-treated wastewater samples) to apply the different methods. Already there are some concerns regarding the manuscript, mainly when discussing the results. | |
The discussion would largely benefit if the authors were to compare with previous papers performed for viruses instead of comparing mainly with bacterial results. There are several other publications in viruses such as KArim et al. 2015 "Propidium monoazide reverse transcriptase PCR and RT-qPCR for detecting infectious enterovirus and norovirus" (Journal of Virological Methods Volume 219, July 2015, Pages 51-61), Leifels et al 2016 "Use of ethidium monoazide and propidium monoazide to determine viral infectivity upon inactivation by heat, UV- exposure and chlorine" (International Journal of Hygiene and Environmental Health Volume 218, Issue 8, November 2015, Pages 686-693), Randazzo et al. 2016 "Evaluation of viability PCR performance for assessing norovirus infectivity in fresh-cut vegetables and irrigation water" (Int J Food Microbiol. 2016 Jul 16;229:1-6) and Monteiro and Santos 2018 "Enzymatic and viability RT‐qPCR assays for evaluation of enterovirus, hepatitis A virus and norovirus inactivation: Implications for public health risk assessment" (J Appl Microbiol. 2018 Apr;124(4):965-976), which the authors can use to benefit the discussion. | The discussion has been modified to include more virus related articles. Reference (such as the ones you kindly recommended) have been added for application of PMA/EMA after incativation with UV light, addition of chlorine and temperature for a variety of DNA and RNA viruses and bacteriophages. Thank you for this comment! |
there are some other issues with the paper. It is not clear how many samples were tested for each wastewater type, were they 24 samples in total for the wastewater samples regardless of the type or were they 24 for each type?). | has been added to the introduction and the material and methods section |
Since the authors explain briefly the concentration of the environmental samples, it is not clear how the secondary concentration with PEG was performed and it is not clear the resuspension buffer, if any was used. | more information added to the text |
The authors should explain, if briefly, the cloning protocol used since they refer to another manuscript. The authors do not describe if they use a one-step or a two-step qPCR for the detection of MNV and if it was a one-step qPCR which kit was applied. | added to material and methods |
On page 7, line 193, the authors state that "Although HAdV-Heim amplicons are only half the length of HAdV-Hernroth" which is disagreement with the numbers presented on table 1 (page 6) the authors should re-write the sentence to be in agreement with Table 1. | line hase been changed accordingly |
Is the legend from the Y axis on figure 1 correct? Shouldn't it be Log10 reduction as stated in the figure's legend? Additionally, Figure 3 and 4 appear in the text before figure 2. Additionally, Figure 4 is upside down. | changed in the text according to reviewers comments (sorry for not spotting the upside-down figure before!) |
It would be relevant to understand why heat and chlorine treatment were able to complete inactivate Adenovirus as measured by cell culture but the same did not happen with UV treatment. | more information and references have been added |
Other interesting result that the authors fail to address is the fact that a lower percentage of positive results were obtained when analyzing sewage pre-UV when compared to sewage post-UV by cell culture, PMA and EMA-ci-qPCR although the qPCR results are similar for both types of sewage | discussion has been extended |
The authors should also check if the % of positive samples for PMA on lines 278/279 before UV treatment is correct (45%). | corrected in the text |
On page 7 the authors have two statements lines 193-195 "Although HAdV-Heim amplicons are only half the length of HAdV-Hernroth (which should statistically influence the number of azo-dye molecules bound to free genomic material), similar differences in the ability to remove non-infectious virions (‘false-positive’ results) were observed." and lines 213 and 215 "Still, they differ in the length and sequence of the targeted area, which most likely influences the likelihood of a successful binding of EMA/PMA and the subsequent formation of the genome dye complex" that seems to contradict each other and the finding the authors have obtained. The authors should rearrange the text. | second statement was removed from the text |
. The authors state that the higher number of negative environmental samples obtained with EMA-PCR compared to cell culture and PMA-PCR was possibly due to the entrance of EMA into competent cells. Besides the controls added in the dye pretreatment section, didn't the authors also tested an additional control where they treated infectious viruses with EMA and PMA? the manuscript would highly benefit from this. | each dye treatment included a "positive process control" where naked nucleic acids of the virus under investigation was added to PMA/EMA to ensure that the incubation and light treatment worked properly. Additionally, the dye treatment included a "negative process control" in cell culture supernatant with know concentrations (based on TCID50) which were treated with EMA and PMA and signal reduction in follow-up qPCR showed to be very low (less then one log). Therefore, It could be suspected that unfortunately these virus stocks were dissimilar to environmental HAdV in regards of their overall "fitness" and their susceptibility of being entered by EMA or PMA. Further research into this would be advised but has not been conducted in the context of this study. |
Reviewer 2 Report
This paper is interesting and the authors have done a good job comparing methods. Some modifications are required before its publication
Specific comments
L17: originating from point and non-point sources. Delete sewage.
L130: this method has poor recovery. The best option is just to simply drop the pH of the sample and then pass through the HA membranes. Extract DNA directly from the membrane.
L133: Who is the manufacturer of the membranes?
L138: How appropriate MNV is to determine virus recovery for HAdV?
L139: The authors should show show the data. These data are important. The recovery after DNA extraction or just the method recovery?
L143: How did you determine the final concentration 0.04mM?
L174: 25 gene copies per L - of what? If it is water, then I would like to known how you calculated this. This is PCR LOD not method LOD. The way the sentence is written it sounds like 25 GC is method LOD. Revise.
L188-192: Redundant. Delete.
Author Response
Thank you very much for your comments and suggestions. Please find the response to each comment in the table below and the revised manuscript text.
This paper is interesting and the authors have done a good job comparing methods. Some modifications are required before its publication | |
Specific comments | |
L17: originating from point and non-point sources. Delete sewage. | deleted in the text |
L130: this method has poor recovery. The best option is just to simply drop the pH of the sample and then pass through the HA membranes. Extract DNA directly from the membrane. | thank you for the advise, we will make sure to include this in future virus recovery tests! |
L133: Who is the manufacturer of the membranes? | added to the text |
L138: How appropriate MNV is to determine virus recovery for HAdV? | citation added; MNV was chosen as a process control for other RNA viruses under investigation in the project this presented work was a part of (the concentration of HAV and Noroviruses was also determined). It is not optimal for HAdV but we decided to continue using it due to the good experience we made with it and as we could cultivate on RAW cells at that time. |
L139: The authors should show show the data. These data are important. The recovery after DNA extraction or just the method recovery? | changed in the text and noted for future projects to include process control and recovery rates into the supplementary, sorry for failing to do so here |
L143: How did you determine the final concentration 0.04mM? | preliminary experiments showed that lower concentrations of dye showed a closer resemblance to cell culture than the 100µM used by Prevost et al, 2016 or Randazzo et al 2016/2018. Again, next time we will include the optimization process in the supplementary material. |
L174: 25 gene copies per L - of what? If it is water, then I would like to known how you calculated this. This is PCR LOD not method LOD. The way the sentence is written it sounds like 25 GC is method LOD. Revise. | changed in the text |
L188-192: Redundant. Delete. | deleted in the text |
Reviewer 3 Report
Recommendation: Publish after minor revisions noted.
Comments:
This manuscript describes the potential of capsid integrity (ci-) qPCR to reduce false-positive results and its application for molecular human adenovirus (HadV) quantification. This manuscript is interesting and results should be published after the authors consider the following minor concerns below.
1. Pg. 3 line 72: Please consider a few sentences to briefly describe the capsid integrity (ci-) qPCR and its advantages for molecular quantification with references.
2. Pg. 8 lines 226-228 : A comparative chart is needed to show sensitivity and specificity.
3. Despite its advantages, the PMA-based approach has known practical and theoretical limitations. Please consider to explain your observation with references.
4. Please consider to include the optimization data for the incubation temperature and duration, as well as the concentration of PMA used.
5. Please consider to include data and discuss false-positive signals due to the penetration of PMA via damaged cell membranes during your experiment.
Author Response
Thank you very much for your comments and suggestions. Please find the response to each comment in the table below and the revised manuscript text.
This manuscript describes the potential of capsid integrity (ci-) qPCR to reduce false-positive results and its application for molecular human adenovirus (HadV) quantification. This manuscript is interesting and results should be published after the authors consider the following minor concerns below. | |
1. Pg. 3 line 72: Please consider a few sentences to briefly describe the capsid integrity (ci-) qPCR and its advantages for molecular quantification with references. | added to the introduction |
2. Pg. 8 lines 226-228 : A comparative chart is needed to show sensitivity and specificity. | reference to the very detailled Rames et al. 2016 Review on this topic has been added |
3. Despite its advantages, the PMA-based approach has known practical and theoretical limitations. Please consider to explain your observation with references. | more information and references on the limitations of PMA/EMA after UV disinfection has been added to the text |
4. Please consider to include the optimization data for the incubation temperature and duration, as well as the concentration of PMA used. | thank you for this comment, we will make sure to include data obtained during the optimization process into future publications. The optimization and validation process was part of a master thesis by a colleague and it is planned to publish it seperately. As we want to sure that she will not encounter any problems due to parts of her work being published before this paper, we did not include the optimization here. |
5. Please consider to include data and discuss false-positive signals due to the penetration of PMA via damaged cell membranes during your experiment. | added to the discussion |
Round 2
Reviewer 1 Report
The manuscript has improved greatly with the changes performed by the authors. However, there are still some minor comments:
- page 9, line 301: it is not clear which samples were spiked and the spiking conditions (concentration of viruses, etc...);
- since the authors removed Figures 3 and 4 from their original place in the text, the reference to these figures has completely disappeared from the text;
- page 9, line 299: the sentence as it is does not make sense. A possible "and" is not correct "showed by [73] and for influenza viruses and [74]...";
- page 11, line 334: change the sentence since it has "the a fraction"
Author Response
Thank you very much for your valuable suggestions and observations, all your remarks have been addressed in the following table and the manuscript has been changed accordingly.
The manuscript has improved greatly with the changes performed by the authors. However, there are still some minor comments:
| |
- page 9, line 301: it is not clear which samples were spiked and the spiking conditions (concentration of viruses, etc...);
| Has been changed in the text and additional information on the concentration of virus added has been added to the material and method section |
- since the authors removed Figures 3 and 4 from their original place in the text, the reference to these figures has completely disappeared from the text;
| References to both figures have been added to the text (sorry for forgetting to update this!) |
- page 9, line 299: the sentence as it is does not make sense. A possible "and" is not correct "showed by [73] and for influenza viruses and [74]...";
| The “and” has been removed |
- page 11, line 334: change the sentence since it has "the a fraction"
| “the” has been removed in the text |